# Teacher Education Interventions on Teacher TPACK: A Meta-Analysis Study

**Yimin Ning [1]** , **Ying Zhou [1],\*** , **Tommy Tanu Wijaya [2],\*** and **Jihe Chen [3]**

1   School of Mathematics and Statistics, Guangxi Normal University, Guilin 541006, China
2   School of Mathematical Sciences, Beijing Normal University, Beijing 100875, China
3   New Century School, Dongguan 523700, China
*   Correspondence: zhouying66@mailbox.gxnu.edu.cn (Y.Z.); 202139130001@mail.bnu.edu.cn (T.T.W.)

**Abstract:** Teacher education is an important strategy for developing teachers' technological pedagogical content knowledge (TPACK). Many schools in the world have incorporated the training into teacher education plans. However, there has been controversy in academic circles concerning the effects of teacher education intervention in promoting the development of teacher TPACK. Therefore, this study used a meta-analysis approach to review the published literature on teacher education programs to determine the impact on TPACK. The results showed that teacher education intervention positively affected TPACK (d = 0.839, *p* < 0.0001). Besides cultural background, experimental participants, types, sample types, intervention durations, differences in measurement methods, intervention types, and learning environments are the reasons for the differences in the effects of the interventions. The research design using random experiments had a significant positive effect on the size, which was significantly higher than that of the quasi-experiment. The longer the duration of teaching intervention, the stronger the improvement effect of teachers' TPACK. There are significant differences in improving TPACK between teaching interventions, and the effect is more obvious. Teacher education intervention has a greater and slightly smaller impact on theoretical and practical knowledge. However, cultural background, experimental participant, sample type, and learning environment have no significant effect on teacher education intervention.

**Keywords:** teacher education intervention; TPACK; meta-analysis

## 1. Introduction

The Technological Pedagogical Content Knowledge (TPACK) framework provides both empirical and theoretical guidance for technology integration in the classroom. The TPACK framework is an important framework for current teacher education. Since the formal introduction of TPACK theory, many studies have recognized the broad appeal and potential of the framework. It is a theoretical basis for developing teachers' understanding of using technology to support student learning constructively and has become one of the frontiers in educational technology [1]. Teachers' TPACK knowledge is not fixed and can be cultivated by designing a specific system of education programs [2]. Since the concept, many teachers' education programs have been developed internationally to cultivate TPACK. Many professional development courses have also been reorganized to promote development [3]. However, the effect of education programs and their substantive impact have always been debated. Some studies believe the program intervention significantly promotes teachers' TPACK [4,5]. Furthermore, the intervention had no significant effect on TPACKs development [6,7]. Are teacher education program interventions effective for TPACK development? Is the moderating effect of experiment type, sample type, and intervention duration significant on the effect of teacher education intervention? This study used the meta-analysis method to quantitatively integrate relevant experimental research conclusions, analyze the influence of different moderator variables on teachers' TPACK

improvement effects, and provide guidance for implementing the program intervention to answer the above question.

## 2. Literature Review

### 2.1. TPACK Theory

Since the concept of technological pedagogical content knowledge, studies have put forward a variety of theoretical frameworks to analyze the connotation from different theoretical orientations. This study grasps the connotation by reviewing the development of the concept.

The conceptualization is mainly derived from Shulman's theoretical framework of technological pedagogical content knowledge (TPACK) [8]. In the early 2000s, adding technical knowledge to Schulman's pedagogical knowledge base was also suggested, and in 2001, Pierson began to use the concept [9]. Niess defines TPCK as a complex of disciplinary, technical, and teaching and learning knowledge development. It studies how technology integration projects affect pre-service teachers' use of technology in teaching practice [10]. According to the Education Association, TPCK is composed of all consonants, which are difficult to pronounce and remember. Therefore, the vowel letter A is added to the abbreviation of TPCK, and the subject teaching knowledge of integrated technology is officially changed to TPACK [11]. In a five-year graduate professional development program, Koehler and Mishra developed the TPACK theoretical framework by engaging students in designing online courses, educational videos, or redesigning existing websites [12]. The TPACK framework proposed by Koehler and Mishra has become a generally accepted framework. It presents the overall framework to understand the complexity of TPACK, but it is difficult to clearly define the internal concepts and the relationship. Angeli and Valanides questioned and explicitly criticized the TPACK superposition view from the perspective of epistemology. The growth of a certain knowledge base (technical knowledge, pedagogical knowledge, or content knowledge) will spontaneously lead to the growth of TPACK [2]. Cox and Graham emphasized the relationship between TPACK and PCK. Furthermore, they questioned the TPACK framework, pointing out the dynamic nature from the perspective of rapid technological changes [13]. Doering et al., explained the dynamic nature of the bidirectional relationship between knowledge and practice [14].

In summary, the following three views of TPACK have developed over time: T as an augmentation of PCK (PCK) [8,15], as a unique and distinct body of knowledge TPCK [9,10], as interactions among the three types of knowledge and their crossover in specific contexts [12]. The most widely recognized TPACK framework consists of eight parts, as seen in Figure 1 [12].

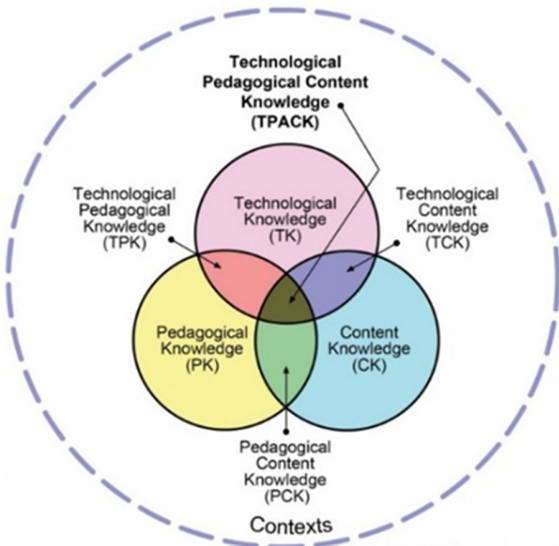

**Figure 1.** TPACK framework and its components.

## 2.2. Teacher Education Interventions

In the context of TPACK, teacher education is an intervention in developing teacher TPACK [16]. Studies have confirmed that teachers' TPACK is affected by individual and teaching factors [17]. By changing teaching strategies and continuously strengthening effective teacher education interventions, it is possible to promote teacher TPACK development. Since the introduction, there have been many ways to help teachers develop TPACK [18], and this includes focusing on learning technology [19] or trying to inject it into other educational courses, such as educational psychology or teaching methods [20].

According to different intervention methods, research on developing teachers' TPACK strategies can be divided into the following three categories: method, tool, and technical intervention [21]. Methods interventions are the most widely used strategies, specifically learning by design [22], scaffolding teaching [23], collaborative learning [24], problem-based teaching [25], case study [26], and game learning [27]. Studies have also attempted to build models such as TPACK-COIR [28], TPACK-COPR [24], and TPACK-IDDIRR [29] during design learning.

Tool intervention tools can be divided into multimedia, including graphics, audio, video, and 2D/3D animation, micro-lectures [30], presentation tools such as Spreadsheets [31], and Web2.0 tools such as WebQuest [32]. Technical intervention can be divided into the following three categories: AI (computer intelligent tutoring system) [24], data collection and analysis software [33], and interactive whiteboard [34]. Many teaching strategies and information technology tools are used in teacher education, and there are diverse combinations.

In order to explore the effect of teacher education on teachers' TPACK intervention, the researchers conducted experimental demonstrations, but the conclusions did not reach a consensus. Most studies agree that teacher education intervention has a significant effect on improving teachers' TPACK. For example, Ching Sing Chai studied the development of TPACK in 78 pre-service teachers under blended learning conditions and found that the integration of information and communication technology in education Both student-centered approaches are new strategies for the development of TPACK for science teachers [35]. Similarly, a study by Irina Lyublinskaya and Nelly Tournaki found that preservice special education teachers had significantly improved TPACK in math and science courses [36], and Karl Wollmann et al. also came to the same conclusion [37]. However, some studies have confirmed that teacher education interventions have no significant effect on improving teachers' TPACK. For example, Fatih Saltan found that online cases significantly improved participants' technical knowledge and technical content knowledge, but only focused on technical knowledge, content knowledge, and technical knowledge. Teaching knowledge is not enough to develop teachers' TPACK [38], and Seong-Won Kim et al. also came to a similar conclusion [39]. These mixed findings suggest that the impact of teacher education interventions on teacher TPACK is unclear; therefore, a systematic approach is needed to review the effects of teacher education interventions. Meta-analysis is a quantitative research method widely used worldwide. It can avoid the biases and deficiencies of traditional research methods to a certain extent and obtain more general and regular research conclusions. Therefore, this study adopts the method of meta-analysis to deeply explore the impact of teacher education on TPACK and provide a reference for improving education.

## 3. Methods and Materials

### 3.1. Literature Search and Screening

#### 3.1.1. Literature Search

The strategy of combining literature retrieval and snowballing was adopted in the first round. This study searched several international databases to ensure the representativeness and comprehensiveness of the literature. The types include journal papers, conference papers, and dissertations. In order to ensure the comprehensiveness of the literature search, the author conducted a comprehensive search on the English and Chinese databases. Literatures were retrieved from databases such as Web of Science, Google Scholar, ProQuest,

and Scopus. The keywords and qualifiers to be searched are set according to the search criteria. The specific screening process and criteria are as follows:

(1) The author's preset search criteria are as follows: the research content is teacher TPACK, so "technological pedagogical content knowledge OR TPACK" is selected as the search term. According to the connotation and structure of TPACK, the core of the TPACK framework is technical knowledge, so we can further filter "technological knowledge OR TK OR technological content knowledge OR TCK OR technological pedagogical knowledge OR TPK" as the search term;

(2) The literature is experimental research, which can provide quantitative data results, so select "experiment" is used as the search term.

(3) The literature is from the field of education, so "education" is selected as the qualifier;

(4) Since TPACK was officially proposed in June 2006, so the time span is determined to be from June 2006 to July 2022;

(5) The age range of the tested teachers, countries, and regions is not limited;

(6) The sources of literature are journals and dissertations, so the type of literature is limited to journals and dissertations.

This study also conducted a secondary search through citation backtracking to ensure comprehensive coverage. The above search strategy was used to obtain 2490 pieces of literature, and a total of 1105 were obtained after excluding the duplicate.

### 3.1.2. Literature Selection Criteria

The inclusion criteria for the literature are as follows:

(1) Research object: The impact of teacher education on TPACK;

(2) Study results: The results of selected studies should show changes in TPACK. This meta-analysis takes teacher education as the independent variable and TPACK as the dependent variable;

(3) Research type: Types of quasi-experimental or randomized experimental studies. During the experiment, students should not be informed of the purpose of the research, and the experimental intervention duration should be more than 1 week. The research findings may be skewed when the study duration is too brief;

(4) Study content: To avoid a disproportionate impact on the overall results, the content of selected studies was reviewed to exclude the same study published in different formats;

(5) Research results: The literature should present clear and complete statistical results. Studies should use standardized tests to measure TPACK and contain sufficient statistical information, including mean, standard deviation, sample size, or t-value, and F-value, to ensure that effect sizes can be calculated.

The exclusion criteria for the literature are:

(1) Research object: Excludes research on the effect of teacher TPACK intervention on teachers whose research topic is not a teacher education program;

(2) Study results: Studies with findings that did not show changes in TPACK were excluded;

(3) Study type: Referring to the criteria of Cheung and Slavin, studies with large differences (effect size > 0.5) were excluded, and random experiments with no pre-test and intervention duration less than one week were excluded [40];

(4) Research content: Exclude the same research published in different formats;

(5) Research results: Literatures without sufficient statistical information were excluded.

### 3.1.3. Literature Screening Process

The first step was retrieval, and a total of 2490 articles were retrieved, followed by three rounds of screening. The second step is the primary screening; the titles are screened, and the documents are imported into EndnoteX9 for screening. A total of 1105 documents were collected after removing duplicates and research published in multiple formats.

The third step is confirmation; after screening the abstracts and excluding irrelevant or non-experimental research on the research topic, a total of 199 papers were obtained. The fourth step is inclusion; the content is reviewed, and 59 papers are finally obtained. Separately, 59 effect sizes were retrieved when many independent samples were included in a single piece of literature (Figure 2).

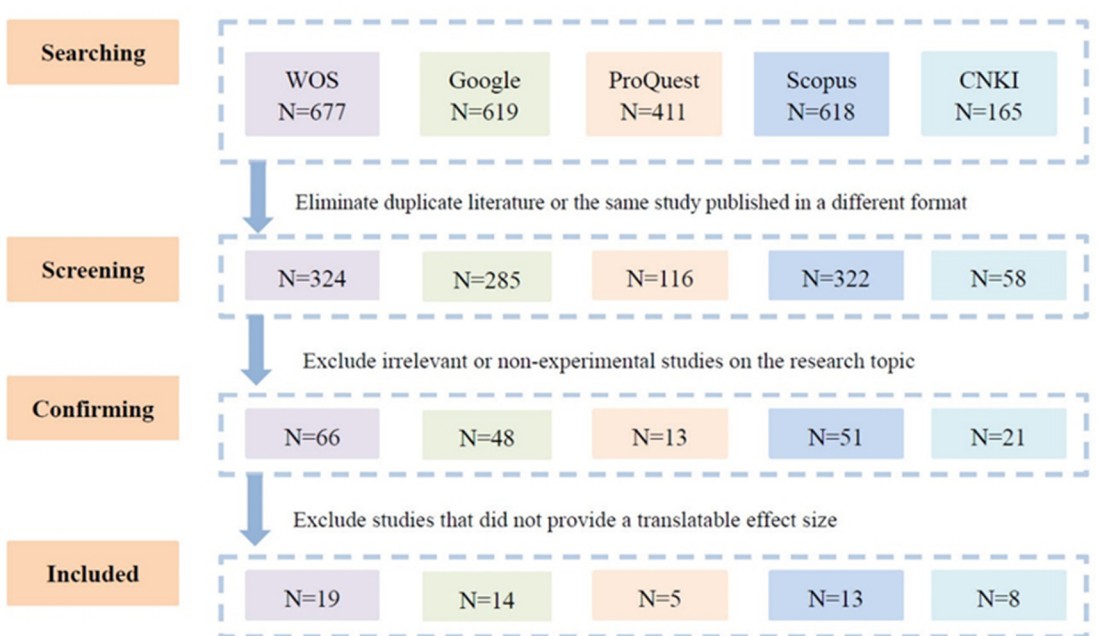

**Figure 2.** Literature screening process.

*3.2. Document Coding*

The purpose of coding is to facilitate moderator analysis and group comparison based on the organization of the literature. The code for this study is as follows:

(1)   Literature information, including author names, publication years, and journal sources (graduation thesis, international conferences, journal papers);

(2)   Cultural background, including eastern and western cultures, coded according to Hofstede's cross-border cultural survey data [41]. Hofstede's cross-cultural analysis model includes the following six dimensions: Power Distance, Uncertainty Avoidance, Individualism/Collectivism, Masculinity/Femininity, Long/Short-Term Orientation, and Indulgence/Restraint. Among them, power distance refers to the unequal distribution of power within an organization. According to Hofstede's cross-border cultural survey data [42], countries with higher power distances such as mainland China, Singapore, Malaysia, Arab countries (such as Kuwait), Indonesia, Turkey, South Korea, and Spain are coded as east. Meanwhile, countries with lower power distances, such as the United States, Germany, and Australia, are coded as the west. Furthermore, when English literature belongs to an eastern country, it is also compiled as East;

(3)   Experimental participants refer to the types of teachers tested, including pre-service and in-service;

(4)   The types of experiments can be divided into random experiments, quasi-experiments, and real experiments according to the control of educational experiments. The independent variables of educational experiments often have comprehensive characteristics, and there is no real experiment in the true sense. Therefore, educational experiments include random experiments and quasi-experiments;

(5) The sample type is divided into the following two types according to the size: the large sample is coded as L, while the small is coded as S. Referring to Chueng and Slavin [40], a sample with a size greater than 250 is a large sample, while less than or equal to 250 is small;

(6) Intervention duration refers to the duration of TPACK training and teaching using the intervention. According to the length of the experimental time, the duration is standardized as 0~3 months, 3~6 months, and more than 6 months;

(7) Measurement methods using Schmidt [43], Chai [20], and Koehler [44] classification hierarchy as measurement tools are coded as standardized. Other adaptations of Schmidt, Chai, and Koehler or a combination of multiple frameworks are coded as self-edited measurements;

(8) Types of interventions, according to the names of teaching interventions reported by the authors, are divided into methods including design learning, scaffolding teaching, case learning, problem-based teaching, game learning; technical comprising of interactive whiteboards, and micro-lectures; tool interventions such as AI, multimedia, digital resources, and robotics;

(9) The types of knowledge are divided into theory and practice. Theoretical courses focus on students' acquisition of concepts, rules, and principles, while practical courses focus on mastery of operational skills;

(10) Teaching environment refers to the environment in which the process is completed. In the context of TPACK, teacher education environments include online, offline, and hybrid [45];

(11) Intervention results, according to the *p* value provided by the literature, the intervention results of teacher education programs are divided into the following two types: improvement and no significant difference;

The coding method is that the first author independently codes, and then the corresponding author checks and proofreads simultaneously after completion. The agreement between the two independent coders was 97%, except for individual data biases, and the specific codes of the 59 studies are shown in Table 1.

**Table 1.** Literature coding results.

| Author (Year) | Cultural Back-Ground | Experimental Participant | Experiment Type | Sample Type | Intervention Duration | Measurement Method | Intervention Type | Knowledge Type | Teaching Environment | Intervention Outcome |
|---|---|---|---|---|---|---|---|---|---|---|
| Seong-Won Kim et al. (2018) [46] | E | P | R | S | 3–6 months | SD | M | In | Off | Improve |
| Eunjung Lee et al. (2019) [47] | E | P | Q | S | 3–6 months | SM | M | Th | Off | Improve |
| Karl Wollmann al. (2022) [37] | W | P | R | L | 3–6 months | SD | M | Th | Off | Improve |
| Julio Cabero et al. (2016) [48] | E | B | Q | L | 3–6 months | SM | M | Th | Off | Improve |
| Joyce Hwee Ling Koh (2018) [7] | E | I | Q | S | 3–6 months | SD | M | Th | On | Improve |
| Seong-Won Kim et al. (2017) [35] | E | P | R | S | 3–6 months | SD | M | In | Off | No significant difference |
| Cevdet Cengiz (2014) [5] | W | P | Q | S | 3–6 months | SM | T | Th | Off | Improve |
| Meng Yew Tee et al. (2011) [49] | E | I | Q | S | 3–6 months | SM | M | Pr | Off | Improve |
| Ching Sing Chai et al. (2010) [50] | E | P | R | L | 0–3 months | SD | M | Th | Off | Improve |
| Insook Han et al. (2013) [4] | E | P | Q | S | 0–3 months | SM | T | In | Off | Improve |
| Ghaida M. Alayyar et al. (2012) [51] | W | P | Q | S | 3–6 months | SM | T | In | Mix | Improve |
| Seong-Won Kim et al. (2016) [39] | E | P | R | S | >6 months | SD | M | Th | Off | Improve |
| Teemu Valtonen et al. (2019) [52] | W | P | Q | S | 3–6 months | SM | M | In | Off | Improve |

**Table 1.** *Cont.*

| Author (Year) | Cultural Back-Ground | Experimental Participant | Experiment Type | Sample Type | Intervention Duration | Measurement Method | Intervention Type | Knowledge Type | Teaching Environment | Intervention Outcome |
|---|---|---|---|---|---|---|---|---|---|---|
| Ching Sing Chai et al. (2011) [53] | E | P | R | L | >6 months | SD | M | Th | Off | Improve |
| Joyce Hwee Ling Koh et al. (2017) [6] | E | I | Q | S | 3–6 months | SM | M | Th | Off | Improve |
| Jacob A. Hall et al. (2019) [54] | W | P | R | S | 3–6 months | SD | M | Th | Mix | Improve |
| Sug Shin, Won et al. (2012) [55] | E | P | R | S | 0–3 months | SD | M | Th | Off | Improve |
| Kyungsik Choi et al. (2019) [56] | E | P | Q | S | 3–6 months | SM | M | Th | Off | Improve |
| Choi Young-mi et al. (2019) [57] | E | P | Q | S | 0–3 months | SM | M | In | Off | Improve |
| Syh-Jong Jang et al. (2012) [34] | E | I | R | L | 0–3 months | SD | T1 | Th | Off | Improve |
| Surattana Adipat (2021) [58] | E | P | R | S | 3–6 months | SD | M | Th | Off | Improve |
| Choi Young-mi et al. (2021) [59] | E | P | R | S | 0–3 months | SM | M | Th | Off | Improve |
| Ching Sing Chai et al. (2020) [60] | E | P | Q | S | 3–6 months | SM | M | Th | Mix | Improve |
| Jennifer R. Banas et al. (2014) [61] | W | P | Q | S | 3–6 months | SM | M | Pr | Off | Improve |
| Ayla Cetin-Dindar et al. (2017) [62] | W | P | Q | S | 0–3 months | SD | E | Th | Off | Improve |
| Douglas D. Agyei et al. (2015) [63] | W | P | Q | S | 0–3 months | SM | M | Pr | Off | Improve |
| Chun-Yen Chang et al. (2012) [64] | E | P | Q | S | 3–6 months | SM | T1 | In | Mix | Improve |
| Park Ye-Rang et al. (2021) [65] | E | P | Q | S | 3–6 months | SM | M | Th | Off | Improve |
| Joo Young-Joo et al. (2012) [66] | E | P | R | S | 0–3 months | SD | M | Th | On | Improve |
| Hyun-Jong Choi et al. (2015) [67] | E | P | Q | S | 3–6 months | SM | M | In | Mix | Improve |
| Arwa Ahmed Abdo Qasem (2016) [33] | E | P | R | S | 0–3 months | SD | M | Th | Mix | Improve |
| Andreas Lachner et al. (2021) [68] | W | P | R | S | 0–3 months | SD | M | Th | Mix | Improve |
| Franziska Zimmermann et al. (2021) [69] | W | P | R | S | 3–6 months | SM | T1 | Th | Off | Improve |
| Aleksandra Kaplon-Schilis (2018) [70] | W | B | Q | S | >6 months | SD | M | Th | Off | Improve |
| Ching Sing Chai et al. (2017) [23] | E | P | Q | L | 3–6 months | SM | M | In | Mix | No significant difference |
| Ping-Han Cheng et al. (2022) [71] | E | P | Q | S | 0–3 months | SM | T | In | Mix | Improve |
| Umit Izgi-Onbasili et al. (2022) [72] | W | P | R | S | 3–6 months | SD | T1 | Th | Off | No significant difference |
| Zheng Zhigao et al. (2019) [73] | E | P | Q | S | 0–3 months | SM | M | In | Mix | No significant difference |
| Yao Liang (2021) [74] | E | P | R | S | 3–6 months | SD | T1 | Pr | Off | Improve |
| Chen Xile (2017) [75] | E | P | R | S | 0–3 months | SD | M | In | Mix | Improve |
| Zhang Mingrui (2021) [76] | E | P | R | S | 3–6 months | SM | M | Pr | Mix | Improve |
| Souphanh Thephavongsa (2019) [77] | E | P | Q | S | >6 months | SM | M | In | Mix | No significant difference |
| Li Yonghan (2021) [78] | E | P | Q | S | 3–6 months | SM | M | In | Off | Improve |
| Fatih Saltan (2017) [38] | W | P | R | S | 0–3 months | SD | T | Th | Mix | No significant difference |
| Liang Cunliang (2015) [79] | E | I | R | S | 3–6 months | SD | T | In | Off | Improve |
| Wang Chunli (2012) [80] | E | P | Q | S | 0–3 months | SM | M | Pr | Off | Improve |
| Dr. Imran Ansari (2019) [81] | E | P | Q | S | 3–6 months | SM | M | Pr | Mix | Improve |
| Kaushal Kumar Bhagat et al. (2017) [82] | E | P | Q | S | 3–6 months | SD | T1 | Th | Mix | Improve |
| Irina Lyublinskaya et al. (2014) [83] | W | P | Q | S | >6 months | SM | T | Th | Off | Improve |
| Jewoong Moon et al. (2022) [84] | W | P | R | S | 3–6 months | SD | M | Pr | Off | Improve |

**Table 1.** *Cont.*

| Author (Year) | Cultural Back-Ground | Experimental Participant | Experiment Type | Sample Type | Intervention Duration | Measurement Method | Intervention Type | Knowledge Type | Teaching Environ-ment | Intervention Outcome |
|---|---|---|---|---|---|---|---|---|---|---|
| Joyce Hwee Ling Koh et al. (2014) [17] | E | B | Q | L | 0–3 months | SM | M | Th | Off | Improve |
| Bian Wu et al. (2021) [85] | E | P | Q | S | 0–3 months | SD | M | Pr | On | Improve |
| Lourdes Meroño et al. (2021) [86] | W | P | Q | L | 3–6 months | SM | T1 | Pr | Off | Improve |
| Erkko Sointu et al. (2016) [87] | W | P | R | S | 3–6 months | SD | T1 | Pr | On | Improve |
| Chew Cheng Meng et al. (2013) [49] | E | P | Q | S | 3–6 months | SM | M | In | Off | Improve |
| Mehmet Barış Horzum (2013) [88] | W | P | R | L | 3–6 months | SD | M | In | Off | Improve |
| Tezcan Kartal et al. (2021) [89] | W | P | Q | S | 0–3 months | SM | M | Th | Off | Improve |
| Hasniza Nordin et al. (2013) [90] | W | P | Q | S | 3–6 months | SM | M | Th | Off | Improve |
| Piret Lehiste (2015) [91] | W | I | Q | S | >6 months | SD | M | In | Off | Improve |

Note: Cultural background—east (E), west (W). Experimental participant—pre-service teacher (P), in-service teachers (I). Experiment Type—Random experiment (R), quasi-experiment(Q). Sample type—large sample (L), small sample (S). Measurement method—standardized test (SD), self-made test (SM). Intervention type—method intervention (M), technical intervention (T), tool intervention (T1). Knowledge type—theoretical (Th), practical (Pr), integrated (In). Teaching environment—online (On), offline (Off), mixed type (Mix).

### 3.3. Literature Quality Assessment

In a meta-analysis, the quality of the included literature affects the final results, and this study refers to Cooper et al.'s literature quality assessment method [92]. It scores the literature quality according to the included literature described in the sample characteristics, experimental design, interventions, measurement tools, and measurement process. Uncertainty receives a score of 0, relative clarity receives a point, and clarity receives a score of 2. A maximum of 10 points can be awarded for a document that is directly related. Two studies were conducted on the evaluation process to ensure the objectivity of the literature quality evaluation results. The score was 0.867 ($p < 0.001$), indicating that the included literature quality met the standard's requirements.

### 3.4. Research Tools

The data analysis tool was the meta-analysis software (Comprehensive Meta-Analysis, CMA). The data used in the meta-analysis includes the number of samples and the mean and standard deviation of the experimental and control groups before and after the test. These raw data are input into the CMA software to generate the effect value of each sample. Since teacher TPACK is a continuous variable, the included literatures are all randomized experiments or quasi-experimental designs to compare differences between or within groups. The sample size of the literature is small, hence this study uses Hedges' as the effect size.

## 4. Analysis of Results

### 4.1. Heterogeneity Test

Due to the heterogeneity between studies, such as cultural background, sample size, intervention measures, and teaching methods, it is necessary to judge the model based on the test results. Heterogeneity is "the differences between all studies that include the same meta-analysis." The purpose of the test, also called the statistical homogeneity test or the consistency test, is to check the consistency results of each independent study. Combinability and heterogeneity testing include a statistical and a graphical method. Commonly used heterogeneity indicators include Q statistic, H statistic, and $I^2$ statistic. The Q statistic is affected by the number of included documents. The H and the $I^2$ statistics are corrected for the degree of freedom (number of documents) of the Q statistic, which will not change with the number of documents included, and the results are more stable and reliable. The heterogeneity test mainly refers to the $I^2$ value [93].

Heterogeneity was further tested by plotting forest plots of fixed-effects models for the 59 studies. The chi-square value is 1807.85, the degree of freedom is 58, and the *p* value is less than 0.01, indicating obvious heterogeneity among the 59 pieces research, as illustrated by the forest plot's results. At the same time, the F value is 92%, verifying the significant heterogeneity among the 59 studies. This is the same result presented by the funnel plot. Hence, further analysis of the sources of heterogeneity is required. Subgroup analysis, meta-regression, or sensitivity analysis were used to explore the source of heterogeneity. After excluding the influence of obvious heterogeneity, a random-effects model was used for meta-analysis.

This study used meta-regression analysis to explore sources of heterogeneity in pooled effects. A meta-regression test was performed with the measurement method as a covariate, $p = 0.000 < 0.05$, I = 93.8%, and R-squared (%) = 0.47, indicating that the measurement method could explain 47% of the heterogeneity. The test was performed on the variables, and $p = 0.0086 < 0.05$, I = 95.61%, and R-squared (%) = 0.26, indicating that the intervention measures can explain 26% of the heterogeneity. Equal covariates can explain more sources of heterogeneity.

### 4.2. Evaluation of Publication Bias

Publication bias is caused by relying on the direction and strength of research findings when selecting papers for publication. The publication has a certain degree of selectivity, and studies with statistically significant positive results are easier or faster to publish. Commonly used detection methods include funnel plots, Egger's, Begg's, and loss of safety factor [94]. Therefore, to fully consider the above impact of publication bias results and ensure the reliability of the meta-analysis, a funnel plot was used to evaluate publication bias.

The funnel plot mainly uses visual observation to identify publication bias. It takes the effect size as the abscissa, the ordinate as the standard error, and the two slashes as the 95% confidence interval. Ideally, the interval and the dispersion obtained by small samples are larger; hence it is often at the bottom of the funnel plot, and the dispersion of large samples is smaller and at the top under normal circumstances. The publication bias of the 59 experimental studies included in the meta-analysis was assessed using CMA software, which was graphed as shown in Figure 3. The test results for the funnel plot in Figure 3 are not symmetrical. This study chose Egger's test slightly more powerful than Begg's and is more sensitive to small samples. The results showed that t = 1.757 > 1.96, p_1, p_2 < 0.05, further indicating the existence of bias. However, through the analysis of the safety factor, the value of the loss of safety factor is 911, which is much larger than "N × 5 + 10". Therefore, publication bias exists to a certain extent, but it is still safe.

**Figure 3.** Funnel plot for publication bias assessment.

### 4.3. Main Effect Test

There is heterogeneity among the initial studies, indicating that in addition to random errors, there are other factors leading to the real difference between the effect sizes of the studies. Therefore, this study uses a random effect model to analyze teacher education (see Table 2). The effect size of 59 pieces of literature as the outcome variable, the combined effect size of a teacher education intervention on teacher TPACK is 0.839. It can be seen that teacher education intervention has a significant positive effect on teacher TPACK.

**Table 2.** Main effects test.

| Model | Effect Size | Effect Value | 95% Confidence Interval | | Two-Tailed Test | | Heterogeneity Test | |
|---|---|---|---|---|---|---|---|---|
| | | | Lower Limit | Upper Limit | Z Value | *p* Value | df (Q) | I-Squared |
| Random effects | 59 | 0.839 | 0.632 | 1.045 | 7.971 | 0.000 | 58 | 96.05018 |

Among the included literature, 22 reported significant, statistically significant effects, and 13 had an effect size greater than 1. The forest map can be drawn using CMA software. See the forest plot for details, as shown in Figure 4.

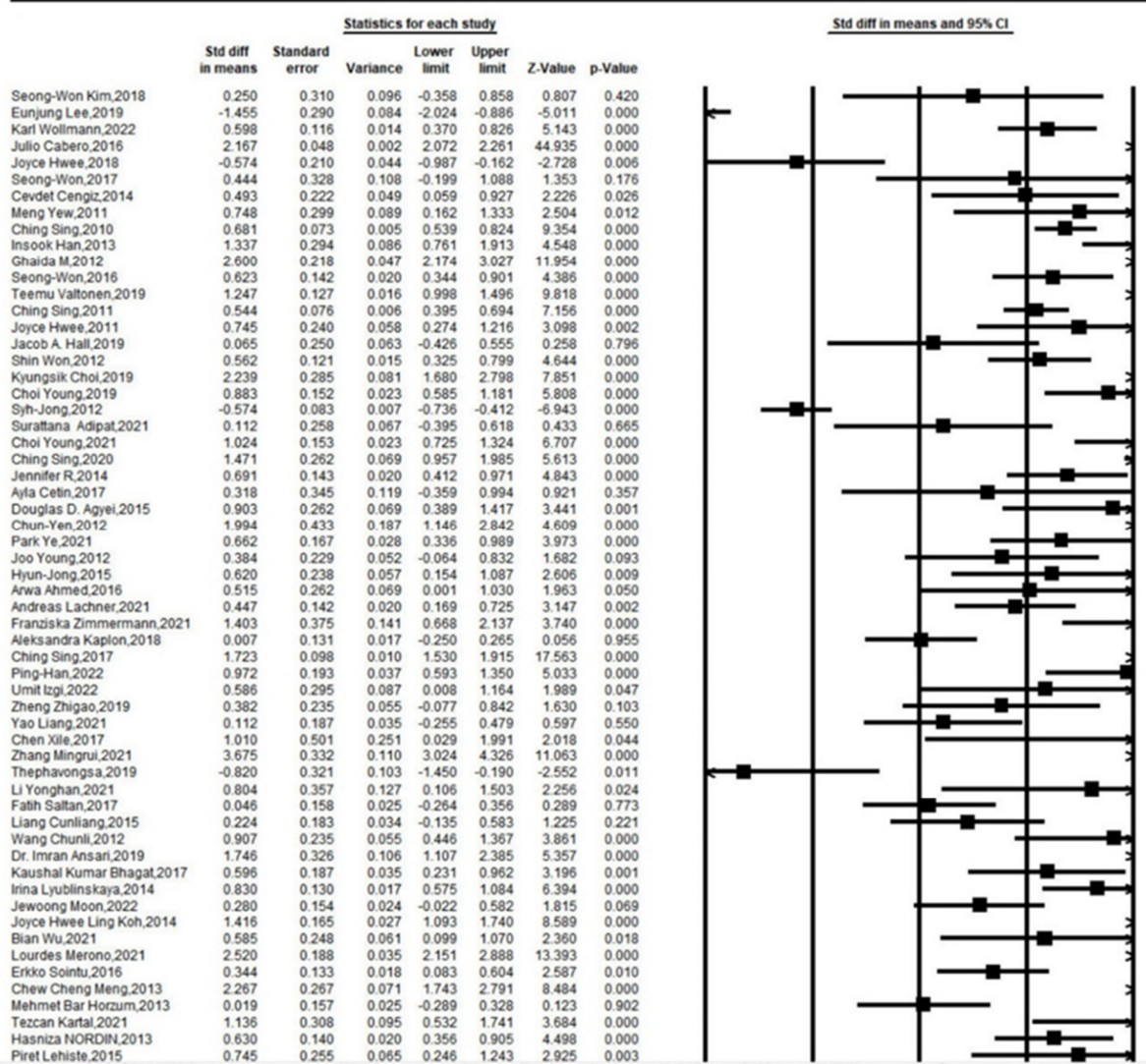

**Figure 4.** Forest map. Note: Mingrui Zhang (2021) [79]. Ghaida M. Alayyar et al. (2012) [52].

Lourdes Meroño et al. (2021) [89]. Chew Cheng Meng et al. (2013) [50]. Kyungsik Choi et al. (2019) [58]. Julio Cabero et al. (2016) [48]. Chun-Yen Chang et al.(2012) [67]. Ching Sing Chai et al. (2017) [23]. Joyce Hwee Ling Koh et al. (2014) [17]. Dr. Imran Ansari (2019) [84]. Ching Sing Chai et al. (2020) [63]. Joyce Hwee Ling Koh et al. (2014) [17]. Franziska Zimmermann et al. (2021) [72]. Insook Han et al. (2013) [4]. Teemu Valtonen et al. (2019) [53]. Tezcan Kartal et al. (2021) [92]. Choi Young-mi et al. (2021) [59]. Ping-Han Cheng et al. (2022) [74]. Xile Chen (2017) [78]. Chunli Wang (2012) [83]. Douglas D. Agyei et al.(2015) [66]. Choi Young-mi et al. (2019) [59]. Irina Lyublinskaya et al. (2014) [86]. Yonghan Li (2021) [81]. Joyce Hwee Ling Koh et al. (2014) [17]. Piret Lehiste (2015) [94]. Meng Yew Tee et al. (2011) [50]. Jennifer R. Banas et al. (2014) [64]. Ching Sing Chai et al. (2010) [51]. Park Ye-Rang et al.(2021) [68]. Hasniza Nordin et al. (2013) [93]. Seong-Won Kim et al. (2016) [39]. Hyun-Jong Choi et al. (2015) [70]. Karl Wollmannet al. (2022) [37]. Kaushal Kumar Bhagat et al. (2017) [85]. Bian Wu et al. (2021) [88]. Umit Izgi-Onbasili et al. (2022) [75]. Sug Shin Won et al. (2012) [57]. Ching Sing Chai et al. (2011) [54]. Cevdet Cengiz (2014) [5]. Andreas Lachner et al. (2021) [71]. Seong-Won Kim et al. (2017) [35]. Joo Young-Joo et al. (2012) [69]. Zhigao Zheng et al.(2019) [76]. Erkko Sointu et al. (2016) [90]. Ayla Cetin-Dindar et al. (2017) [65]. Jewoong Moon et al.(2022) [87]. Cunliang Liang (2015) [82]. Liang Yao (2021) [77]. Surattana Adipat (2021) [61]. Jacob A. Hall et al. (2019) [56]. Fatih Saltan (2017) [38]. Mehmet Barış Horzum (2013) [91]. Aleksandra Kaplon-Schilis (2018) [73]. Joyce Hwee Ling Koh(2018) [49]. Syh-Jong Jang et al. (2012) [60]. Thephavongsa (2019) [80]. Eunjung Lee et al. (2019) [47].

### *4.4. Moderating Effect Test*

The above heterogeneity test results show that each literature's effect sizes has significant statistical significance variability. This is necessary to analyze the source of this variability through the moderation effect test. To understand the impact of a teaching intervention on students' deep learning, this study uses cultural background, experimental participant, experimental type, sample type, intervention durations, measurement method, intervention type, and teaching environment as moderator variables to test the effect. The results showed that cultural background, experimental participants, experimental types, sample types, intervention durations, measurement methods, intervention types, and learning environments were the reasons for the differences in the effect sizes of each study.

#### 4.4.1. Moderating Effect Test of Different Cultural Backgrounds

Table 3 presents the combined effect size obtained by statistics to analyze the differences in the effect of teacher education intervention in different cultural backgrounds. The cultural background has no significant moderating effect on teacher education intervention ($Q = 0.603$, $p > 0.05$). From the perspective of different cultural backgrounds, the combined effect size of Eastern and Western cultures is 0.900 and 0.730, respectively. The *p*-values of both groups were less than 0.001, reaching a statistically significant level. The combined effect value of Western culture is between 0.5 and 0.8, indicating that under the background of Western culture, teacher education intervention has a moderate promotion effect on teacher TPACK. In contrast, the combined effect value of Eastern culture is greater than 0.8, implying that teachers' educational intervention significantly affected TPACK. Under different cultural backgrounds, the TPACK level of teachers is different. However, the programs are all based on the TPACK model, and there is a great similarity in training goals and design that can significantly improve the TPACK of teachers in the region. Therefore, the moderating effect of cultural background is not significant. Eastern countries are more inclined to develop TPACK through a separate program. In contrast, most Western countries integrate it into the teacher education curriculum through the entire teacher education process [95]. Therefore, compared to short-term teacher education programs, courses that integrate information technology have a more profound impact on teacher TPACK development.

**Table 3.** Moderating effect test.

| Moderating Variable | Categories | Effect Size | Effect Value | 95% Confidence Interval | | Two-Tailed Test | | Heterogeneity Test | |
|---|---|---|---|---|---|---|---|---|---|
| | | | | Lower Limit | Upper Limit | Z Value | *p* Value | df (Q) | I-Squared |
| Cultural background | East | 38 | 0.900 | 0.642 | 1.157 | 6.852 | 0.000 | | |
| | West | 21 | 0.730 | 0.389 | 1.072 | 4.191 | 0.000 | | |
| Total between | | | | | | | | 0.603 | 0.437 |
| experimental participants | Pre-service teachers | 50 | 0.864 | 0.670 | 1.059 | 8.726 | 0.000 | | |
| | In-service teachers | 6 | 0.399 | −0.156 | 0.955 | 1.408 | 0.159 | | |
| | Mixed type | 3 | 1.204 | 0.444 | 1.964 | 3.106 | 0.002 | | |
| Total between | | | | | | | | 3.355 | 0.187 |
| Experimental type | Quasi-experiment | 35 | 0.580 | 0.299 | 0.861 | 4.041 | 0.000 | | |
| | Random experiment | 24 | 1.026 | 0.791 | 1.260 | 8.574 | 0.000 | | |
| Total between | | | | | | | | 5.696 | 0.017 |
| Sample type | Small sample | 50 | 0.109 | 0.575 | 1.001 | 7.247 | 0.000 | | |
| | Large sample | 9 | 0.246 | 0.652 | 1.616 | 4.614 | 0.000 | | |
| Total between | | | | | | | | 1.658 | 0.198 |
| Intervention durations | 0–3 months | 20 | 0.177 | 0.345 | 1.039 | 3.906 | 0.000 | | |
| | 3–6 months | 33 | 0.139 | 0.689 | 1.232 | 6.937 | 0.000 | | |
| | >6 months | 6 | 0.321 | 0.100 | 1.360 | 2.271 | 0.023 | | |
| Total between | | | | | | | | 1.574 | 0.455 |
| Measurement method | Self-made test | 32 | 0.367 | 0.139 | 0.595 | 3.160 | 0.002 | | |
| | Standardized test | 27 | 1.250 | 1.037 | 1.463 | 11.521 | 0.000 | | |
| Total between | | | | | | | | 3.863 | 0.000 |
| Intervention type | Method Intervention | 43 | 1.013 | 0.785 | 1.242 | 8.685 | 0.000 | | |
| | Tool intervention | 9 | 0.586 | 0.083 | 1.088 | 2.285 | 0.022 | | |
| | Technical intervention | 7 | 0.155 | −0.406 | 0.715 | 0.541 | 0.588 | | |
| Total between | | | | | | | | 8.936 | 0.011 |
| Knowledge type | Theoretical | 29 | 1.013 | 0.023 | 0.396 | 4.551 | 0.000 | | |
| | Practical | 11 | 0.586 | 0.063 | 0.619 | 4.431 | 0.000 | | |
| | Integrated | 19 | 0.155 | 0.037 | 0.553 | 4.823 | 0.000 | | |
| Total between | | | | | | | | 2.270 | 0.321 |
| Teaching environment | Online | 4 | 0.896 | 0.216 | 1.576 | 2.582 | 0.000 | | |
| | Offline | 38 | 0.808 | 0.583 | 1.032 | 7.054 | 0.010 | | |
| | Mixed type | 17 | 0.912 | 0.569 | 1.254 | 5.215 | 0.000 | | |
| Total between | | | | | | | | 0.272 | 0.873 |

### 4.4.2. Moderating Effect Test of Different Experimental Participant

To analyze the differences in the intervention effect of teacher education on different participants, Table 3 presents the combined effect size obtained by statistics. The experimental participant has no significant moderating effect on the effect of teacher education intervention (Q = 3.355, *p* > 0.05). The combined effect size of pre-service teachers was 0.864, and 1.204 for pre-service and in-service teachers. The *p*-values of both groups were less than 0.001, reaching a statistically significant level. Former teachers have a significant positive promotion effect, followed by mixing pre-service and in-service teachers. The main reason for this result is that pre-service teachers are in the development stage of the TPACK level, and the effect of the intervention is significant. In contrast, the teaching methods of in-service teachers are relatively fixed, and their TPACK level is stable. Hence, teacher education intervention is not obvious. The number of studies on in-service teachers is small due to the sample size, and the impact needs to be further explored [96].

### 4.4.3. Moderating Effect Test of Different Experimental Types

Table 3 presents the combined effect size obtained by statistics to analyze the differences in the effects of teacher education intervention under different experimental types. The experimental type significantly moderates teacher education intervention (Q = 5.696, *p* < 0.05). From the perspective of different experimental types, the combined effect size of the quasi and random experiments is 0.580 and 1.026, respectively. The *p*-values of both groups were less than 0.001, reaching the level of statistical significance. This shows significant differences in teacher education intervention's effect under different experiments.

The research design using randomized experiments positively impacts the effective value, which is higher than the quasi-experiment. The size of the effect value is highly correlated with the quality of the experimental design.

### 4.4.4. Moderating Effect Test for Different Sample Types

To analyze the differences in the effect of teacher education intervention under different experimental types, Table 3 presents the combined effect size obtained by statistics. The moderating effect of sample type on the effect of teacher education intervention was insignificant (Q = 1.658, $p > 0.05$). The small and large samples' combined effect sizes were 0.109 and 0.246, respectively. The $p$-values were all less than 0.001, reaching a statistically significant level. The sample type of citations has no significant moderating effect on the effect of teacher education intervention. Therefore, there is no significant difference in moderating effects between small and large samples. The number of studies with large samples is too small, which is limited by the study's sample size, and the impact needs to be further explored.

### 4.4.5. Moderating Effect Test of Different Intervention Durations

Table 3 presents the combined effect size obtained by statistics to analyze the differences in the effect of teacher education intervention under different intervention durations. The intervention durations had no significant moderating effect on the teacher education intervention (Q = 1.574, $p > 0.05$). The combined effect size of 0–3 and 3–6 months was 0.177 and 0.177. The combined effect size of more than 6 months was 0.023, and the $p$-values of the three groups were all less than 0.05, reaching a statistically significant level. The effect of a teacher education intervention on teacher TPACK is small when the intervention duration is less than 6 months. The combined effect value of the research with an intervention duration of more than 6 months is the highest, above other subgroups. This phenomenon may be because the formation of teacher TPACK is a long-term process, the intervention duration in citations is mostly within 6 months, and the impact is insignificant. Internal TPACK knowledge structures are not yet established when the trial time is too short, making the major impact problematic. The meta-analysis results showed that the longer the intervention duration, the more significant the effect of a teacher education intervention on TPACK [44]. However, a more detailed analysis could not be conducted due to the lack of studies over the last 6 months [97].

### 4.4.6. Moderating Effect Test of Different Measurement Methods

Table 3 presents the combined effect size obtained by statistics to analyze the differences in the effect of teacher education intervention under different measurement methods. The measurement method significantly moderates the teacher education intervention (Q = 3.863, $p < 0.05$). From the perspective of the measurement method, the standard test has a greater impact on the effect of the intervention. The significant effect of standardized tests indicates that using tools can more accurately explore the relationship between teacher educational interventions and the effect of teacher TPACK.

### 4.4.7. Moderating Effect Test of Different Intervention Types

Table 3 exhibits the combined effect size obtained by statistics to analyze the differences in the effect of teacher education intervention under different types of intervention. The type of intervention has a significant moderating effect on teacher education intervention (Q = 8.936, $p < 0.05$). In terms of intervention measures, method has the best effect, followed by tool, while the important technical intervention has the most effect. Pure technical intervention is not as effective as a systematic method, which also shows that the learning and application of TPACK should emphasize the technology and information environment. of "teaching and learning theories" and methods [98]. Technology interventions may be used with other teaching strategies to facilitate teachers' TPACK development [99]. In addition, since the number of studies involved is relatively small, and the complex

nature of the TPACK development process suggests that it is necessary to use only method intervention for teaching. Therefore, follow-up research needs to focus on technology intervention in teachers' TPACK.

### 4.4.8. Moderating Effect Test of Different Knowledge Types

To analyze the differences in the effect of teacher education intervention, Table 3 presents the combined effect size obtained by statistics. From the perspective of the between-group effect, Q = 2.270 and $p$ = 0.321 > 0.05. Therefore, there are no significant differences in the effect of a teacher education intervention on the learning effect of different knowledge types. The combined effect value of theoretical and practical knowledge is 1.013 and 0.586. Comprehensive knowledge was 0.155, and both were significant at 0.001 ($p$ < 0.001). Teacher education intervention has a significant positive effect on teachers' TPACK. Specifically, this has a greater and smaller impact on the learning effect of theoretical and practical knowledge, respectively. This is because the transformation process from theory to practice is complex, and teacher education programs are more direct and do not provide enough guidance for the practice. Even though teachers use technology in training and show evidence of TPACK, their pedagogical methods are not entirely consistent with those emphasized in the professional development process [100].

### 4.4.9. Moderating Effect Test of Different Teaching Environments

Table 3 presents the combined effect size obtained by statistics to analyze the differences in the effect of teacher education intervention under different types of intervention. With the development of technology, the cooperative learning environment is not limited to offline classroom learning. Furthermore, it is also possible to carry out cooperative learning in online and blended environments. The teaching environment significantly modifies teacher education intervention (Q = 0.272, $p$ > 0.05) in different teaching places. Teacher education intervention can positively affect TPACK, which is consistent with existing research conclusions [101]. The effect of the intervention in the blended learning environment has the highest effect size. Therefore, blended and online teaching environments have a greater impact on teachers' TPACK [102]. This is because the development of TPACK requires a technology-rich environment. The presentation of TPACK theoretical knowledge or the application of TPACK in teaching practice relies on the support of technology. Therefore, online teaching is more effective than offline. Blended teaching integrates online teaching due to its positive effect [33].

## 5. Discussion and Inspiration

### 5.1. Teacher Education Intervention Has a Significant Impact on Teacher TPACK

Using a meta-analysis approach, this study conducted a statistical analysis of 59 randomized experimental or quasi-experimental studies on the effects of teacher education interventions on TPACK. The common effect value of teacher education intervention on the overall impact was 0.839, consistent with the conclusion of Lyublinskaya et al. [83]. The collected research can support the conclusion that teacher education intervention can significantly improve TPACK. However, the design and implementation should fully consider the intervention measures, knowledge types, and teaching environment. This result can be summarized in the literature included in the analysis as follows: First, teacher education programs combine new technology knowledge with content and teaching, from only contacting existing theoretical knowledge and technology. Resources are turned to practice and application to improve teachers' TPACK level [103]. Second, with the development of information technology, teacher education has integrated online and offline teaching environments [104].

### 5.2. The Effect of Moderator Variables on Teacher Education Intervention

This study also analyzed the reasons for the differences in the results of each study through the moderation effect test. The included moderator variables can be divided into

the following two categories: One is variables related to the research design, including cultural background, experimental type, sample type, intervention duration, and measurement method. The others are variables related to the teaching intervention, including experimental participants, intervention types, knowledge types, and teaching methods. The results of the study are as follows:

The moderating effect of cultural background on teacher education intervention is insignificant, and the effect is better under the background of oriental culture. Even though most of the existing literature discusses the relationship between cultural backgrounds, there are significant differences in teachers' TPACK ability [105]. However, cultural factors' influence on teacher education intervention's effect is less discussed. The most likely explanation is that there are also significant cultural and educational differences between Eastern and Western countries, preventing all studies from being included in these two categories of the moderation analysis. The relationship between cultural background and teacher education intervention's effect needs to be clarified further.

The type of experiment has a significant moderating effect on teacher education intervention, and the design using randomized experiments has a significant positive impact. The effect size of teacher education intervention depends on the outcome measurement method used and the rigor of the experimental design. It is highly correlated with the quality of the experimental design [106].

The moderating effect of sample type on the teacher education intervention is insignificant, and the value of large and small samples is not high. This can be understood from the small effect of the two class variables and the *p*-value indicating the share of this moderator variable. The result is statistically unreasonable because effect sizes are susceptible to outliers with small sample sizes, and results from larger studies are given more weight. This bias in the heterogeneity analysis may be because too little literature was included. This led to the interference of outliers in teacher education interventions and the inability to draw reliable conclusions [107]. Since the number of studies varies, the difference in effect size does not simply mean that large samples are better than small. There are various reasons for this difference, such as variation in sample sizes, experimental participants, and interventions.

The intervention duration has no significant moderating effect on the teacher education intervention. The longer the intervention duration, the more significant the effect of a teacher education intervention on TPACK. This finding suggests that efficiency may be a useful variable to consider when conducting research in the future. Teacher TPACK training has a certain "life cycle" because effective improvement needs to go through a series of development stages, including cognition, acceptance, adaptation, exploration, and improvement [108]. Therefore, the intervention duration is one factor determining the effectiveness of teachers' educational interventions. The internal TPACK structure of teachers has not been formed, and it is difficult to significantly impact their TPACK levels.

The measurement method significantly moderates teacher education intervention, and the standard test has a greater influence. Studies using standardized tests showed that teacher education interventions had a greater effect on TPACK than self-administered tests. Consistent with existing research conclusions [109], the number of studies involved in the self-made test is relatively large. Therefore, the effect size is likely to be affected by other factors. The effect size of teacher education intervention depends on the rigor of the experimental design, which is highly correlated with the measurement of TPACK.

The moderator variables related to the research design have been discussed above. Regarding the variables related to the teaching intervention, the results of the study are as follows:

The experimental participants had no significant moderating effect on the teacher education intervention. Therefore, pre-service and in-service teacher groups benefited equally from technology. Teacher education programs had the greatest impact on pre-service teacher TPACK of all study characteristics relevant to the sample. From the existing empirical research, technology is more attractive to pre-service teachers. They are more

willing to receive TPACK training and make more effort to learn. For technical knowledge learning, the TPACK level tends to be mature and stable [96]. A separate TPACK education program was used to improve the teaching effect for pre-service teachers.

The type of intervention has a significant moderating effect on the effect of teacher education intervention. As the most direct measure, the technical intervention has no significant effect. Method intervention is based on analyzing the psychological mechanism formed by the teachers' TPACK. Technology and tools are key factors, but the current integration has not kept pace with the rapid changes in the quality and quantity of information technology [110]. Due to the limitation of teachers' cognitive level of TPACK, technical intervention still needs the majority of educational studies to improve and develop this important intervention technology in future theoretical research and teaching practice.

In teacher education intervention, the learning effect of theoretical and practical knowledge is more influential and slightly smaller. The teacher education program provides a platform to learn this knowledge. Furthermore, teachers are more sensitive to theoretical knowledge but fail to establish an organic connection. Some can use TPACK in learning, but technical knowledge is rarely combined in real classroom teaching for various reasons. There is still a gap between theoretical and practical teaching [111]. For this reason, different types of knowledge points should be different when designing TPACK teaching. More background knowledge of the organizer can be designed before the theoretical class. The important and difficult points need to be internalized and explained in the classroom. For practical knowledge, all operations can be arranged before the class. The class focuses on learning deeper skills, and design targets can give full play to students' subjective initiative and creative exploration activities.

From the perspective of the teaching environment, the effect size is greater than 0.8, indicating that the intervention had a significant positive impact on teacher TPACK. A blended teaching environment that combines the advantages of online and offline learning is significantly more effective than purely online and traditional offline. The main reason for this phenomenon is the enhanced situation creation and interaction [33].

## 6. Conclusions and Implications

### 6.1. Conclusions

Through the method of meta-analysis, 59 studies were comprehensively sorted out on the effect of teacher education on teachers' TPACK intervention over the past ten years and objectively analyzed and evaluated the effect of the intervention. This study analyzes and discusses the effect of a teacher education intervention on TPACK and the differences under the influence of different moderator variables. In conclusion, the comprehensive effect value of teacher education intervention on TPACK is 0.839. Therefore, the teacher education intervention has a positive promoting effect. From the perspective of moderating effect, the research design using randomized experiments significantly affects the effect size, which is significantly higher than that of quasi-experiments. The longer the duration of teaching intervention, the stronger the improvement effect of teachers' TPACK. There are significant differences in improving TPACK between teaching interventions, and the effect is more obvious. Teacher education intervention has a greater impact on the learning effect of theoretical knowledge and a slightly smaller impact on practical knowledge. However, cultural background, experimental participant, sample type, and learning environment have no significant moderating effect. The meta-analysis results affirmed the importance of teacher education intervention in developing TPACK and its positive role. Education has different intervention cycles, intervention strategies, and knowledge types. There are differences in the impact of effects, and no region, institution, or school can require all education programs to adopt a unified TPACK model.

### 6.2. Limitations and Future Research

Regarding the limitations, the analysis of moderator variables based on a small number of studies should be interpreted cautiously and should not lead to strong inferences. Some

moderator variables involve a small number of studies, resulting in one of the subgroups containing little research literature. Second, some moderator variables (e.g., sample type and intervention duration) may be manipulated inappropriately. This is because the dichotomous criteria for determining specific subgroups may be inaccurate, and such manipulation may increase the analysis's variability, affecting the study's results. Third, the heterogeneity in determining the impact size of teacher education interventions in the meta-analysis suggests that there may be other methodological and non-methodological moderators. Future research should discover these variables moderating the effects of teacher education interventions. This will improve our understanding of the relationship between teacher education interventions and their pre-and post-variables.

Even though the meta-analysis revealed the effect of a teacher education intervention on the development of TPACK, the specific mechanism of action and practical application are not clearly described. This study surveyed schools, institutions, and related teachers to comprehensively understand the intervention's impact on TPACK development.

**Author Contributions:** Conceptualization, Y.N. and T.T.W.; methodology, Y.N.; validation, J.C. and Y.Z.; formal analysis, Y.N. and J.C.; writing—original draft preparation, Y.N.; writing—review and editing, Y.N., T.T.W., and J.C.; visualization, Y.N.; supervision, Y.Z.; funding acquisition, Y.N. All authors have read and agreed to the published version of the manuscript.

**Funding:** This research was funded by the 2021 Guangxi Postgraduate Education Innovation Plan Project "Research on Mathematical Higher-Order Thinking Evaluation Based on Cognitive Maps" (YCSW2021102) and "reform of degree and postgraduate education in Guangxi (JGY2022053)".

**Institutional Review Board Statement:** Not applicable.

**Informed Consent Statement:** Not applicable.

**Data Availability Statement:** Not applicable.

**Acknowledgments:** This research was funded by the postgraduate education innovation project in Guangxi (YJSCX2021103) and reform of degree and postgraduate education in Guangxi (JGY2022053), and thanks to all who participated in this research.

**Conflicts of Interest:** The authors declare no conflict of interest.

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
