# Peer review of "Teacher Education Interventions on Teacher TPACK: A Meta-Analysis Study"

_sustainability, doi:10.3390/su141811791_

Round 1

Reviewer 1 Report

Dear Authors,

First of all, I want to congratulate you on a thorough result and a wonderful publication. I am sure that it will be interesting for colleagues. I had a positive impression after reading the materials, but at the same time, I would like to emphasize some issues that will give the article more clarity.

1. It is necessary to indicate clear dates (month, year) regarding the search interval

2. It is worth making an annex with information about at least one search string for some database

3. It is necessary to provide more information about how such a search string was created (how keywords were selected, whether they were systematized, what algorithm or approach was used for this), whether the search string was tested before performing a data search, etc.

4. What were the criteria for excluding publication?

5. I think it is extremely important to note separately that the literature search was conducted in English and Chinese. It is worth noting, also about other languages, if such literature was analyzed

6. Since there are systematic reviews written about teacher TPACK, it is worth analyzing this literature in the introduction.  This will make it possible to point out even more clearly the existing gaps and the importance of writing exactly this meta-analysis

Kind regards,

Reviewer 2 Report

The reviewer has read carefully the manuscript and stated the following recommendations next to the evaluation above:

- please do not use abbreviations (i.e., e.g., "TPACK", etc.) in the abstract without any explanation or without giving the meaning;

- references from the time interval 2020 and 2022 (2023) are totally missing, please supplement them;

- the DOI numbers are missing from the reference list, please supplement them;

- in Table 1, the references are missing (the reviewer means the numbers in brackets);

- please give the references related to all the figures (objects): if they are not your own work, it is compulsory; if they are your own, please indicate it in the title of the figures (objects), e.g., "own made drawing" or "own made drawing based on xxx ref.";

- how did the authors choose the 59 experimental studies for their research? Why these 59 pieces? Why not other ones?

- please give the answer to the question: is it correct to calculate the parameters with 3 or more digit(s) accuracy in the case you investigated?

- can you improve the manuscript with better visualization of your results (more than tables and incorporated data)? The reviewer recommends preparing, e.g., summarizing figures, charts, etc.

- the manuscript needs English language proofreading, please use native English proofreader;

-

Reviewer 3 Report

This interesting metanalysis study analyzes and discusses the effect of teacher education interventions on TPACK and the differences under the influence of different moderator variables. The results affirmed the importance of teacher education intervention in developing TPACK and its positive role.

The study is generally written in a systematic and understandable way. Nevertheless, further proofreading of the text would help international readers. It is suggested to the authors pay a little more attention to the following comments:

  1)      Authors use many abbreviations without analyzing them the first time they use them. Some of the abbreviations may be taken for granted in the authors’ scientific community, but the article is aimed at an audience of diverse backgrounds, so they should analyze the abbreviation the first time used. Especially for TPACK, which is the central concept of the study, it should be analyzed both in the abstract and the first time it is used in the text.

  2)      I am trying to understand what meaning the authors give to the term "experimental objects" but I am not succeeding. It is mentioned three times in the text, without elaboration. Probably other terminology with the same meaning is used and should be homogenized.

  3)      In the sentence "The most widely recognized TPACK framework consists of eight parts" (lines 76-77) a reference should be added, as in Figure 1 the source should be added.

  4)      I do not understand the meaning of the sentence "Therefore, studies need to grasp the effect of teacher education on TPACK education" (lines 100-101). Perhaps it should become "Therefore, studies need to grasp the effect of teacher education on TPACK", otherwise it would be good to reword or explain.

  5)      The sentence "The foreign language is technological pedagogical content knowledge ..." (lines 113-4) should be reworded. Possible translation: 'In English, the keywords used are: technological pedagogical content knowledge ...'. Similarly, the sentence "In Chinese, the keyword is TPACK OR technical 118 knowledge OR TK OR technical content knowledge OR TCK OR technical teaching 119 knowledge OR TPK OR pre-test OR post-test OR control group OR experiment." (lines 117-120) could be replaced with the shorter "Similar keywords were used for the literature search in Chinese".

  6)      Regarding the Cultural Background (lines 181-7), more details should be provided. A simple reference to Hofstede's study is not sufficient, as Hofstede analyses several indicators. It should at least explain what is meant by 'higher power distance' so that we can understand Spain's integration into the East.

  7)      Regarding The types of experiments, the authors while stating "The independent variables of educational experiments often have comprehensive characteristics, and there is no real experiment in the true sense. Therefore, educational experiments include pre-experiment and quasi-experiment" (lines 190-3), then in Table 1 and the note below it (line 225) refer to random experiment. The random experiment is also referred to throughout the text thereafter. I believe that this contradiction should be considered.

  8)      I'm not sure that the term “experimental period” is the right one. I think using a term related to the duration of the teaching intervention would be more appropriate.

  9)      Finally, I think that the studies that were reviewed should be mentioned somewhere, I mean a full reference, perhaps in an appendix. Maybe the authors already do it, but I have not been able to access the 2 appendices.

Round 2

Reviewer 1 Report

Thank you for the answers and detailed explanations. 

Kind regards,

Author Response

Dear Editors and Reviewers:

Thank you for your letter and for the reviewers’ comments concerning our manuscript entitled “Teacher Education Interventions on Teacher TPACK: A Meta-Analysis Study”  (ID: sustainability-1902653). Those comments are all valuable and very helpful for revising and improving our paper and the important guiding significance to our research. We have studied comments carefully and have made the correction which we hope meet with approval. 

Reviewer 2 Report

The authors prepared my requested corrections, they revised their manuscript well. I recommend the publishing of the paper in the chosen Special Issue.

Author Response

(The authors gave the same response as above.)

Reviewer 3 Report

The authors' responses, at least as recorded in their response letter, seem to be more or less adequate and in line with my comments and suggestions. 

However, the layout of the text this time has serious problems and I can't quite understand the changes that have been made throughout the text. I see text replacements with words stuck together and I also see duplicate versions of terminology, the pre-existing terminology, and the change I requested (e.g. interventiondurationexperimental period) in multiple places in the text. I understand that the authors made changes under time pressure, but unfortunately, the result is a text that is not in publishable form.  I believe the authors should read it once more and make the necessary corrections, this time without trusting the automation of the word processor.

Author Response

Dear Editors and Reviewers:

Thank you for your letter and for the reviewers’ comments concerning our manuscript entitled “Teacher Education Interventions on Teacher TPACK: A Meta-Analysis Study”  (ID: sustainability-1902653). Those comments are all valuable and very helpful for revising and improving our paper and the important guiding significance to our research. We have studied comments carefully and have made the correction which we hope meet with approval. The main correction in the paper and the responses to the reviewer’s comments are as following:

Several statements that we made were more ambiguous than intended, and we have adjusted the text to be clearer.

We appreciate for editors/reviewers’ warm work earnestly and hope that the correction will meet with approval.

Once again, thank you very much for the comments and suggestions.